# Transport Properties in Multicomponent Systems Containing Cyclodextrins and Nickel Ions

**DOI:** 10.3390/ijms25084328

**Published:** 2024-04-13

**Authors:** Sónia I. G. Fangaia, Daniela S. A. Silva, Ana Messias, Pedro M. G. Nicolau, Artur J. M. Valente, M. Melia Rodrigo, Ana C. F. Ribeiro

**Affiliations:** 1Faculty of Medicine, Institute of Implantology and Prosthodontics, University of Coimbra, 3000-075 Coimbra, Portugal; sfangaia@fmed.uc.pt (S.I.G.F.); ana.messias@uc.pt (A.M.); pgnicolau@mail.telepac.pt (P.M.G.N.); 2Center for Innovation and Research in Oral Sciences (CIROS), Faculty of Medicine, University of Coimbra, 3000-075 Coimbra, Portugal; 3CQC, Department of Chemistry, University of Coimbra, 3004-535 Coimbra, Portugal; danielasilva178@gmail.com (D.S.A.S.); avalente@ci.uc.pt (A.J.M.V.); 4Center of Mechanical Engineering Materials and Processes (CEMMPRE), Departamento de Engenharia Mecânica, University of Coimbra, 3030-788 Coimbra, Portugal; 5Universidad de Alcalá, Departamento de Química Analítica, Química Física e Ingeniería Química, 28805 Alcalá de Henares, Spain; mmelia.rodrigo@uah.es

**Keywords:** cyclodextrins, diffusion, multicomponent systems, nickel ions, transport properties

## Abstract

In this work, we propose a comprehensive experimental study of the diffusion of nickel ions in combination with different cyclodextrins as carrier molecules for enhanced solubility and facilitated transport. For this, ternary mutual diffusion coefficients measured by Taylor dispersion method are reported for aqueous solutions containing nickel salts and different cyclodextrins (that is, α-CD, β-CD, and γ-CD) at 298.15 K. A combination of Taylor dispersion and other methods, such as UV-vis spectroscopy, will be used to obtain complementary information on these systems. The determination of the physicochemical properties of these salts with CDs in aqueous solution provides information that allows us to understand solute–solvent interactions, and gives a significant contribution to understanding the mechanisms underlying diffusional transport in aqueous solutions, and, consequently, to mitigating the potential toxicity associated with these metal ions. For example, using mutual diffusion data, it is possible to estimate the number of moles of each ion transported per mole of the cyclodextrin driven by its own concentration gradient.

## 1. Introduction

Cyclodextrins (CDs) are cyclic oligomers composed of glucopyranose units interconnected through α-1,4 glycosidic bonds. Among the most frequently found CDs, the α-, β-, and γ-cyclodextrin variants stand out, consisting, respectively, of six, seven, and eight glucose units. These molecules are soluble in water and have a distinct hydrophobic cavity [1,2].

The most notable feature of cyclodextrins is their distinct ability to form inclusion complexes, both in the solution and in the solid state, in which each guest molecule is enveloped by the hydrophobic environment of the CD cavity. Such complexes are called guest–host complexes [3]. Cyclodextrins have the ability to form complexes with inorganic salts and cations, mainly stable metal ions, which makes them beneficial in several industries, such as pharmaceuticals, cosmetics, textiles, and food [3,4,5]. In aqueous solutions, cyclodextrin molecules tend to incorporate water molecules into their molecular structure, just as metallic salts can adopt the form of neutral hydrates or ionic species. Cyclodextrins have multiple hydroxyl groups, primary and secondary, which can offer coordination sites suitable for binding with metal ions, forming covalent bonds in basic pH environments, where hydroxyl groups can be deprotonated and act as nucleophiles [4].

Due to the characteristics already described and the versatility of cyclodextrins, this study evaluates their application in mitigating the potential toxicological effects of nickel ions released from intra-oral devices.

Dental alloys containing nickel have been widely used for years due to their mechanical properties [6]. Ni-Cr alloys have been applied in the manufacture of crowns and bridges, although nowadays they have fallen into disuse in fixed prosthodontics [7]. Nickel–titanium alloys, also known as Nitinol, are actually widely used in components of orthodontic appliances due to their effectiveness and durability [8,9]. It is also important to highlight that due to the longevity of conventional orthodontic treatments, patients are exposed for many months to the presence of orthodontic metallic components in the oral cavity, which promotes the occurrence of corrosion and wear [9,10,11], and is susceptible to the eventual release of metallic ions from them [8,12,13]. It is essential to note that prolonged exposure to a small amount of this transition metal can affect various cellular functions and also trigger cytotoxic and mutagenic effects [14,15,16,17]. The genetic effects induced by nickel have also been reported in the literature, including damage to DNA and inhibition of enzymes involved in its repair [18,19].

Cyclodextrins have been explored and applied in the production of elixirs and mouthwashes [20,21,22,23,24]. Their wide use in the pharmaceutical field is due to factors, such as low toxicity and safety for human use, low immunogenicity, good cost-effectiveness, and availability. In addition, they are not subject to microbial resistance, increase the stability and solubility of drugs while improving their absorption, mask unpleasant tastes or unwanted odors of certain compounds, regulate the controlled release of drugs, mitigate local and systemic toxicity, improve permeability across biological barriers [3,25,26,27,28,29], and mitigate local and systemic toxicity [22,30].

The evaluation of the diffusion behavior of nickel, in the form of nickel salts in the presence of α-, β-, and γ-cyclodextrins using the Taylor dispersion technique, allows for conclusions to be drawn about the effect of these carbohydrates on the diffusion of this salt, as well as has the potential to provide the scientific and technological community with some values of important parameters in transport processes in solutions. These data may prove useful in the formulations of mouthwashes and mouth rinses.

## 2. Results

### 2.1. Diffusion Measurements

Table 1 shows the pH values for 0.001 mol dm^−3^ NiCl_2_ solutions without and with cyclodextrins (*α*-CD, *β*-CD, or *γ*-CD).

From Table 1, it is observed that for solutions containing NiCl_2_ at 0.001 mol dm^−3^, the pH is around 6.00. At these pH values, we can say that the nickel ion species are predominantly in the non-hydrolysed form, that is, as free Ni^2+^ ions [31].

Table 2 summarizes the average values of the ternary diffusion coefficients *D*_ik_ (i ≠ k) for solutions of different compositions and concentrations for three aqueous systems, involving NiCl_2_ and cyclodextrins (α-CD, β-CD, and γ-CD).

These parameters were calculated by fitting equation (Equation (1)) to the dispersion curves; the number of replicas is always greater than four.

At the limiting situations of X_1_ = 0 and X_1_ = 1, D_11_ values correspond, respectively, to the tracer diffusion coefficient of NiCl_2_ in CDs and the binary mutual diffusion coefficient of aqueous NiCl_2_ at 0.001 mol dm^−3^. Regarding the last D_11_ values of NiCl_2_ for all solutions at X_1_ = 1, reasonable agreement is observed between them and the binary diffusion coefficients previously reported in previous studies (that is, D = 1.048 × 10^−9^ m^2^ s^−1^) [32,33].

In addition, from Table 2, we can see that the cross-diffusion coefficients D_21_ values are close to zero when X_1_ equals one. This means that the salt concentration gradient cannot produce a coupled flux of CD, and therefore the cross coefficients D_21_ must be zero. For the remaining molar fractions (X_1_ = 0 and X_1_ = 0.5), it is also possible to consider them as null, considering the experimental error. This result is likely since the mobilities of the free CD species and the potential aggregates of NiCl_2_ and CDs are similar.

However, it is important to highlight that D_12_ < 0; that is, the concentration gradient of cyclodextrins (α-CD, β-CD, and γ-CD) produces coupled counter current flows of this salt, which assume relevance when X_1_ = 1. In other words, the presence of CD in trace concentration affects the diffusion behavior of nickel ions by the presence of coupled counter current transport of NiCl_2_ from regions of lower to higher concentration of the CD solute.

### 2.2. UV-Vis Spectroscopy Measurements

A calibration curve was initially prepared based on the absorbances at a wavelength of 721 nm, obtaining a linear adjustment up to an Ni-(II) concentration of 350 mmol dm^−3^, with an *R*^2^ value of 0.9991, and a molar absorption coefficient of 2.00 × 10^−6^ mol^−1^ dm^3^ cm^−1^, with the statistical parameters of the validated curve in Table 3.

The interaction between NiCl_2_, and α-CD and γ-CD was also evaluated using UV-vis spectroscopy. Regarding β-cyclodextrin, due to its low solubility in aqueous solutions, it was not possible to carry out this study under the present experimental conditions.

Figure 1 shows the absorption spectra of aqueous NiCl_2_ solutions in the presence of γ-CD.

## 3. Discussion

### 3.1. Hydrolysis of Nickel Ion and Interactions between Cyclodextrins and Nickel Ion as Seen by Diffusion Coefficients Measurements

These observations recorded in the present work can be interpreted through two subsequent phenomena: the hydrolysis of nickel ions and the interaction between NiCl_2_ and cyclodextrin molecules, resulting in the formation of complexes in solution.

Since aqueous solutions of nickel chloride demonstrate a slightly acidic character (as recorded in Table 1), when such salts diffuse into water, the H_3_O^+^ ions resulting from the hydrolysis of nickel ions must first diffuse into the nickel ions, which presents less mobility. This results in the generation of a counter current flow of hydrochloric acid in addition to the main flow of partially hydrolysed nickel chloride. In more explicit terms, given that the H_3_O^+^ ion (λ_m_ = 76.30 × 10^−4^ Ω^−1^ m^2^ mol^−1^) has substantially higher mobility than the Cl^−^ ion (λ_m_ = 349.70 Ω^−1^ m^2^ mol^−1^), a strong electric field is generated. This electric field has the effect of slowing down the H_3_O^+^ ions that drive the considerable counter current flows of Ni-(II) species in a free state or associated with CD molecules. Consequently, this scenario can justify obtaining *D*_12_ values below zero.

The formation of supramolecular aggregates between Ni^2+^ ions and CD molecules can also occur. This process leads to a reduction in free nickel ions. To compensate for this loss, a counter current flow of this salt occurs.

Considering the hypothesis of the formation of a 1:1 supramolecular complex between the Ni^2+^ cation and CD (Equation (1)), and taking into account the values recorded in Table 4 relating to the limiting diffusion coefficients of species in free and complexed states, it is possible to estimate the values of the equilibrium constant, *K*, according to the expression provided by equation (Equation (2)):(1)$${{\mathrm{Ni}}^{2+}}_{\left(\mathrm{aq}\right)}+{\mathrm{CD}}_{\left(\mathrm{aq}\right)}⇆{\mathrm{Ni}}^{2+}-\;\mathrm{CD}\;(\mathrm{aq})$$
(2)$$K\;=\frac{C_{({\mathrm{Ni}}^{2+}-\;\mathrm{CD})}}{C_{{\mathrm{Ni}}^{2+}}\;C_{\mathrm{CD}}}$$

The diffusion coefficients of the Ni^2+^-CD complexes are estimated using the Stokes–Einstein approximation. According to this approach, the diffusion coefficient of a species in solution is inversely proportional to its hydrodynamic radius, and, therefore, inversely proportional to the cube root of its molecular volume (Equation (3)).
*D* = (*D*_NiCl2_^−3^ + *D*_CD_^−3^) ^−1/3^
(3)

The values of the equilibrium constants for the complexed species (Ni^2+^-α-CD, Ni^2+^-β-CD, and Ni^2+^-γ-CD) were calculated, resulting in values equivalent to 270 mol^−1^ dm^3^, 205 mol^−1^ dm^3^, and 125 mol^−1^ dm^3^, respectively. These values suggest a formation of the complexes between the nickel ion and CDs; thus, it is possible to state that small amounts of Ni-(II) and CD molecules can be transported as Ni^2+^-CD complexes.

Information regarding coupled diffusion can also be deduced from the values calculated for the *D*_12_/*D*_22_ ratio (as shown in Table 5). The negative values of this ratio allow us to infer that one mole of α-CD (or β-CD) and γ-CD in the diffusion process counter transports 0.4 and 0.3 moles of NiCl_2_.

### 3.2. Interaction of NiCl_2_ and Cyclodextrins Analyzed Using UV-Vis Spectroscopy Measurements

Analyzing Figure 1, an isosbestic point at a wavelength of 500 nm appears evident. In the shorter wavelength range, absorbance increases as the concentration increases, while in the longer wavelength range, absorbance decreases as the concentration increases. This pattern culminated in the formation of the isosbestic point at 500 nm.

The presence of an isosbestic point during an assay between a metal ion and a target molecule constitutes substantial evidence that only two predominant species are present, having a constant overall concentration. Based on these observations, it appears that the complex 1:1 Ni^2+^-γ-CD is formed (Equation (4)) and has a maximum absorption of 500 nm.
Ni^2+^_(aq)_ + γ-CD _(aq)_ ⇆ Ni^2+^ − γ-CD(4)

The association constant, *K*a, can be estimated according to the Benesi–Hildebrand equation. (Equation (5)) [36],
(5)$$\frac{1}{∆A_{i}}=\frac{1}{∆A}+\frac{1}{∆AKa}\frac{1}{C_{\mathrm{CD}}^{i}}$$
where Δ*A = A*_i_ − *A_o_*, representing *A_o_*, and *A*_i_ the absorbance of NiCl_2_ in the absence of the guest (γ-CD) and the absorbance recorded in the presence of the added guest, $C_{\mathrm{CD}}^{i}$ is the concentration of added cyclodextrin, and Ka is the association constant.

Considering this model (Equation (5)), the association constants between NiCl_2_ com γ-CD and NiCl_2_ with α-CD were estimated (that is, *K*a = 77.5 mol^−1^ dm^3^ and *K*a = 4.82 mol^−1^ dm^3^). This last result indicates that the formation of the complex between the Ni^2+^ ion and α-cyclodextrin is neither particularly favored nor stable under the present experimental conditions. The interaction between these species is relatively weak, suggesting that complex formation may not be as efficient as in cases with higher association constants.

## 4. Materials and Methods

### 4.1. Materials

Table 6 describes all reagents used as received in the present work: α-cyclodextrin, β-cyclodextrin, and γ-cyclodextrin, and nickel chloride. All of these chemicals were used without further purification, but they were stored under low pressure in a desiccator over silica gel.

The solutions were prepared in calibrated volumetric glass flasks, using as solvent, ultrapure water (Millipore, Darmstadt, Germany, Milli-Q Advantage A10, specific resistance = 1.82 × 10^5^ Ω m, at 298.15 K). The weighing was done using a Radwag AS 220C2 balance, with an accuracy of ±0.0001 g.

### 4.2. Methods

#### 4.2.1. pH Measurements

The pH measurements of solutions were carried out with a radiometer pH meter PHM 240 with an Ingold U457-K7pH conjugated electrode; pH was measured in fresh solutions, and the electrode was calibrated immediately before each experimental set of solutions using IUPAC recommended pH 4, 7, and 10 buffers (Table 1). From the pH meter calibration, a zero pH of 6.11 ± 0.03 and sensitivity higher than 98.7% were obtained. All solutions were freshly prepared at 298.15 K and degassed by sonication for about 60 min before each experiment.

#### 4.2.2. Taylor Dispersion Method

The Taylor dispersion technique for measuring mutual diffusion coefficients is well described in the literature [37,38]; consequently, its details are not included in this manuscript. We only summarize some relevant points regarding the method and the equipment. Basically, dispersion profiles were generated by injecting, at the start of each run, 0.063 mL of solution using a 6-port Teflon injection valve (Rheodyne, Bensheim, Germany, model 5020) into a laminar carrier stream of slightly different composition at the entrance to a Teflon capillary dispersion tube of a 3048.0 (±0.1) cm length, and internal radius 0.03220 ± (0.00003) cm. This tube and the injection valve were kept at 298.15 (±0.01) K in an air thermostat. This tube and the injection valve were kept at 298.15 (±0.01) K in an air thermostat chamber. The air inside the thermostat chamber was circulated with the help of a fan and the temperature was kept constant by using a relay connected to a digital thermometer. The broadened distribution of the disperse samples was monitored at the tube outlet by a differential refractometer (Waters, Milford, MA, USA, model 2410). The refractometer output voltage, *V*(*t*), was measured at 5 s intervals by a digital voltmeter (Agilent, Santa Clara, CA, USA, 34401 A).

The dispersion profiles for these ternary solutions NiCl_2_ plus cyclodextrins were analyzed by fitting the following equation (Equation (6)).
(6)$$V\left(t\right)=V_{0}+V_{1}+V_{max}{\left(t_{R}/t\right)}^{1/2}\left[W_{1}exp\left(-\frac{12D_{1}{\left(t-t_{R}\right)}^{2}}{r^{2}t}\right)+(1-W_{1})exp\left(-\frac{12D_{2}{\left(t-t_{R}\right)}^{2}}{r^{2}}\right)\right]$$
to obtain the values of the different *D*_ik_ coefficients. In Equation (6), *D_i_* (*i*: 1 or 2) represents the eigenvalues of the matrix of the ternary *D*_ik_ coefficients; *V*_0_, *V*_1_, and *V*_max_ are the baseline voltage, the baseline slope, and the peak high, respectively; and *W*_1_ and (1 − *W*_1_) are normalized pre-exponential factors.

Diffusion of mixed aqueous solutions of NiCl_2_ (1) plus different cyclodextrins (α, β, or γ-CDs) can be described by the following ternary diffusion equations (Equations (7) and (8))
*J*_1_ = −*D*_11_∇*c*_1_ − *D*_12_∇*c*_2_
(7)
*J*_2_ = −*D*_21_∇*c*_1_ − *D*_22_∇*c*_2_
(8)
where *J*_1_ and *J*_2_ are the molar fluxes of NiCl_2_ (1) and cyclodextrins (α, β, γ) (2) driven by the concentration gradients ∇*c*_1_ and ∇*c*_2_ of each solute 1 and 2, respectively. Main diffusion coefficients *D*_11_ and *D*_22_ give each solute’s flux driven by its concentration gradient. Cross-diffusion coefficients give the coupled flux of each solute driven by a concentration gradient in the other solute. A positive *D*_ik_ cross-coefficient (i ≠ k) indicates the co-current coupled transport of solute I from regions of higher to lower concentrations of solute k. On the other hand, a negative *D*_ik_ coefficient indicates the counter-current coupled transport of solute i from regions of lower to higher concentration of solute k.

#### 4.2.3. Ultraviolet-Visible Spectroscopy

Ultraviolet-visible (UV-vis) spectroscopy measurements were performed using a Shimadzu UV-2600i spectrophotometer, covering the wavelength range between 400 and 800 nm. Analyses were conducted using a quartz cuvette with an optical path length of 1 cm. A concentrated stock solution of nickel chloride hexahydrate in milli-Q water was prepared to construct the calibration curve. Between 10 and 12 standard solutions were prepared in a range of intermediate concentrations, by direct dilution of the stock solution with milli-Q water. The previous solutions were then analyzed using UV-vis spectroscopy.

The desired curve was obtained by representing the absorbances or luminescence intensities read as a function of concentration. It was statistically validated, and its parameters were used to determine the analytical thresholds.

Studies were conducted on the interactions between nickel chloride and different analytes (α, β, and γ-CD). A similar procedure was followed for each analyte. Initially, a concentrated stock solution of nickel chloride hexahydrate was prepared, from which intermediate concentration solutions (usually between 10 mM and 50 mM) were created by dilution in milli-Q water. Next, each compound was weighed and placed in separate sample bottles to which 10 mL of the previously prepared stock solution was added. The calibration curve was obtained by using a set of aqueous nickel chloride solutions prepared by dilution of a stock solution.

## 5. Conclusions

The impact of oligosaccharides (α-cyclodextrin, β-cyclodextrin, and γ-cyclodextrin) on the diffusion of aqueous nickel chloride solutions was investigated. The interaction between the aforementioned compounds was evaluated by measuring the intermolecular diffusion coefficients at 298.15 K. The results revealed a significant interaction between the nickel ion with cyclodextrins, which resulted in obtaining negative values in the secondary diffusion coefficients *D*12. It was also possible to conclude that one mole of CD in the diffusion process transports up to 0.4 moles of NiCl_2_. Support for this interaction between nickel ions and γ-cyclodextrin comes from UV-vis spectroscopy. The presence of an isosbestic point in the absorption spectra indicated a possible association between nickel ions and cyclodextrin (*K*a = 77.5 mol^−1^ dm^3^). In conclusion, this work contributed significantly to the understanding of the mechanisms underlying diffusion transport in aqueous solutions, and, consequently, to the mitigation of the potential toxicity associated with these metal ions. Therefore, it is plausible to propose that incorporating these macrocycle compounds (CDs) in mouthwash formulation could reduce the inherent toxicity associated with metal ions. We consider that the results obtained may be of significant use to the scientific and technological community who are involved in the investigation of these metals, and have clinical implications inherent to their use.

## Figures and Tables

**Figure 1 ijms-25-04328-f001:**
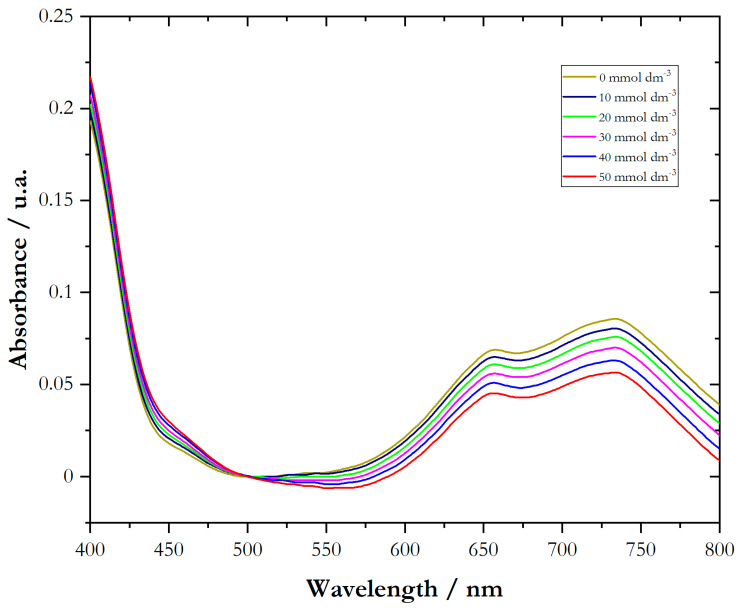
UV-vis absorption spectra of γ-CD in NiCl_2_ (40 mmol dm^−3^). The range of CD concentrations is 10 to 50 mmol dm^−3^.

**Table 1 ijms-25-04328-t001:** pH ^(a)^ measurements for 0.001 mol dm^−3^ NiCl_2_ solutions without and with cyclodextrins (i.e., *α*-CD, *β*-CD, or *γ*-CD at 0.001 mol dm^−3^).

Aqueous System	pH
NiCl_2_	6.08
NiCl_2/_*α*-CD	6.20
NiCl_2/_*β*-CD	5.98
NiCl_2/_*γ*-CD	6.14
*α*-CD	6.40
β-CD	6.00
γ-CD	5.95

^(a)^ From pH meter calibration results with a zero pH of 6.21 ± 0.03 and a sensitivity higher than 98.7% at 298.15 K.

**Table 2 ijms-25-04328-t002:** Ternary diffusion coefficients (*D*_11_, *D*_12_, *D*_21_, *D*_22_ ^(a)^) of aqueous NiCl_2_ (*C*_1_) + CDs (*C*_2_) solutions.

*C*_1_ ^(b)^	*C*_2_ ^(b)^	*X*_1_ ^(c)^	*D*_11_ ± S_D_ ^(d)^	*D*_12_ ± S_D_ ^(d)^	*D*_21_ ± S_D_ ^(d)^	*D*_22_ ± S_D_ ^(d)^
NiCl_2_ (component 1) + α-CD (component 2)						
0.000	0.001	0.00	1.376 ± 0.015	0.069 ± 0.020	−0.002 ± 0.005	0.376 ± 0.002
0.0005	0.0005	0.50	1.158 ± 0.001	0.001 ± 0.003	0.020 ± 0.006	0.369 ± 0.001
0.001	0.000	1.00	1.103 ± 0.001	−0.169 ± 0.052	0.005 ± 0.002	0.393 ± 0.000
NiCl_2_ (component 1) + β-CD (component 2)						
0.000	0.001	0.00	1.372 ± 0.036	0.009 ± 0.002	−0.028 ± 0.005	0.361 ± 0.001
0.0005	0.0005	0.50	1.123 ± 0.011	0.086 ± 0.005	0.006 ± 0.001	0.324 ± 0.003
0.001	0.000	1.00	1.100 ± 0.001	−0.135 ± 0.011	0.003 ± 0.001	0.369 ± 0.001
NiCl_2_ (component 1) + γ-CD (component 2)						
0.000	0.001	0.00	1.312 ± 0.009	0.096 ± 0.014	−0.017 ± 0.010	0.280 ± 0.011
0.0005	0.0005	0.50	1.192 ± 0.024	−0.013 ± 0.002	0.009 ± 0.005	0.357 ± 0.003
0.001	0.000	1.00	1.106 ± 0.000	−0.101 ± 0.021	0.003 ± 0.002	0.351 ± 0.000

^(a)^ Average results from 8 tests (n = 8). ^(b)^ *C*_1_ e *C*_2_ in mol dm^−3^. ^(c)^ *X*_1_ = *C*_1_(*C*_1_ + *C*_2_) represents the the mole fraction of solute NiCl_2_. ^(d)^ *D*_ij_ ± S*_D_* in 10^−9^ m^2^ s^−1^ at *T* = 298.15 K.

**Table 3 ijms-25-04328-t003:** Statistical parameters of the calibration curve based on the absorbances at a wavelength, *γ***,** of 721 nm, for 350 mmol dm^−3^ NiCl_2_ solutions.

*λ*/nm	*b*_1_ (σ(b_1_))/dm^3^ mmol^−1^	*R* ^2^ ^(a)^	*ε*/mol^−1^ dm^3^ cm^−1^ ^(b)^	LOD/mmol dm^−3 (c)^	LOQ/mmol dm^−3 (d)^
721	(0.00200) ±(0.00002)	0.9991	2.00 × 10^−6^	1.60	4.86

^(a)^ Coefficient of determination. ^(b)^ Molar absorption coefficient. ^(c)^ LOD: detection limit. ^(d)^ LOQ: limit of quantification.

**Table 4 ijms-25-04328-t004:** Diffusion coefficients of species, *D*_S_, at 298.15 K.

Species	*D*_s_ (10^−9^ m^2^ s^−1^)
NiCl_2_	1.150 ^(a)^
α-CD	0.361 ^(a)^
β-CD	0.358 ^(a)^
γ-CD	0.239 ^(a)^
NiCl_2_—α-CD	0.367 ^(b)^
NiCl_2_—β-CD	0.354 ^(b)^
NiCl_2_—γ-CD	0.238 ^(b)^

^(a)^ These values are estimates of the diffusion coefficients of free species in water [34,35]. ^(b)^ Equation (3).

**Table 5 ijms-25-04328-t005:** Estimation of the moles of NiCl_2_ transported by each mole of α-CD, β-CD, and γ-CD obtained from *D*_ij_ data presented in Table 2.

Aqueous System	*D*_12_/*D*_22_
NiCl_2_—α-CD	−0.430
NiCl_2_—β-CD	−0.366
NiCl_2_—γ-CD	−0.288

**Table 6 ijms-25-04328-t006:** Sample description.

Chemical Name	Source	CAS Number	Mass Fraction Purity ^(a)^
Nickel chloride hexahydrate	Panreac	7791-13-1	>0.98
α-cyclodextrin ^(a)^	Sigma-Aldrich ^(d)^	10016-20-3	≥0.98
β-cyclodextrin ^(b)^	Sigma-Aldrich ^(d)^	7585-39-9	>0.97
γ-cyclodextrin ^(c)^	Sigma-Aldrich ^(d)^	17465-86-0	≥0.98
H_2_O	Millipore-Q water (ρ = 1.82 × 10^5^ Ω m at 298.15 K)	7732–18-5	

^(a)^ α-cyclodextrin with water mass fraction 0.14. ^(b)^ β-cyclodextrin with water mass fraction 0.13. ^(c)^ γ-cyclodextrin with water mass fraction 0.10. ^(d)^ The mass fraction purity is on water-free basis; these data are provided by the suppliers.

## Data Availability

Data are contained within the article.

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
