# Peer review of "Transport Properties in Multicomponent Systems Containing Cyclodextrins and Nickel Ions"

_ijms, 2024, doi:10.3390/ijms25084328_

Round 1

Reviewer 1 Report

Comments and Suggestions for Authors

The presented work entitled "Transport properties in multicomponent systems containing cyclodextrins and nickel ions" is a very interesting study concerning the experimental investigation of the diffusion process of nickel ions in combination with various cyclodextrins. This work significantly contributes to understanding the underlying mechanisms of diffusive transport in aqueous solutions and, consequently, aims to mitigate the potential toxicity associated with these metal ions. The authors demonstrate the potential application of their findings to mouthwash fluids, which could reduce the natural toxicity associated with metal ions. The authors believe that the results obtained may have significant scientific and technological implications for the community engaged in the research of these metals and the clinical implications inherently associated with their use.

After reviewing the presented work, I have several questions for the authors, and I would appreciate their response:

  1. The authors state that the measurement was performed at a temperature of 298.15K. How was this temperature controlled, as there is no information provided in the text?
  2. In Figure 1, I would suggest changing the light blue color to a more distinct one as the graph is not very clear.
  3. In Table 2, I suggest removing the empty row from row 110.
  4. The authors show the water content in the investigated cyclodextrins. In Table 6, did they calculate how this content could affect the results obtained?
  5. In the final conclusions, the authors mention the application of these studies to mouthwash fluids. Have such studies been conducted before? If so, please provide any relevant literature references.

Author Response

Reply to Reviewers’ comments

Reviewer #1

The presented work entitled "Transport properties in multicomponent systems containing cyclodextrins and nickel ions" is a very interesting study concerning the experimental investigation of the diffusion process of nickel ions in combination with various cyclodextrins. This work significantly contributes to understanding the underlying mechanisms of diffusive transport in aqueous solutions and, consequently, aims to mitigate the potential toxicity associated with these metal ions. The authors demonstrate the potential application of their findings to mouthwash fluids, which could reduce the natural toxicity associated with metal ions. The authors believe that the results obtained may have significant scientific and technological implications for the community engaged in the research of these metals and the clinical implications inherently associated with their use.

We are grateful for these positive comments.

After reviewing the presented work, I have several questions for the authors, and I would appreciate their response:

  1. The authors state that the measurement was performed at a temperature of 298.15K. How was this temperature controlled, as there is no information provided in the text?

We agree with the referee and, consequently, we have inserted additional information in section 4.2.2 Taylor dispersion method. That is:

“This tube and the injection valve were kept at 298.15 (±0.01) K in an air thermostat chamber. The air inside the thermostat chamber is circulated with the help of a fan and the temperature is kept constant by using a relay, connected to a digital thermometer”.

  1. In Figure 1, I would suggest changing the light blue color to a more distinct one as the graph is not very clear.

We agree with the referee and, consequently, we have changed the Figure 1 accordingly.

  1. In Table 2, I suggest removing the empty row from row 110.

The suggested modification was carried out.

  1. The authors show the water content in the investigated cyclodextrins. In Table 6, did they calculate how this content could affect the results obtained?

The concentrations of the all cyclodextrins used were corrected taking to account the mass fraction purity on the water-free basis. In addition, the concentrations of cyclodextrins were corrected for the water content indicated in Table 6. Notably, excluding this quantity would lower solution concentrations and increase D values by less than 1%.

  1. In the final conclusions, the authors mention the application of these studies to mouthwash fluids. Have such studies been conducted before? If so, please provide any relevant literature references.

The authors of this work have conducted similar studies before, which were reported in References 22 and 30. For example, these studies demonstrated that β-CD can interact with vanadium ions. These ions are formed due to the tribocorrosion that Ti-6Al-4V prosthetic devices are subjected to.

Comments and Suggestions for Authors

Reviewer 2 Report

Comments and Suggestions for Authors

The manuscript reports on interesting results on how the cyclodextrins dissolved in mouthwashes can influence the dissolution of nickel from dental alloys and its potential toxicity. It is recommended to publish after minor corrections.

It is suggested to cite the important recent review on the dental applications of cyclodextrins: Braga, S.S. Cyclodextrins as Multi-Functional Ingredients in Dentistry. Pharmaceutics 2023, 15, 2251. https://doi.org/10.3390/pharmaceutics15092251

Give the water content of alpha- and gamma-cyclodextrin, too. (Table 6)

Some typos to correct:

Mind the font size (e.g. lines 88-95)

Line 77: mouthwashes and mouthwashes? (rephrase this sentence)

Line 133: Table 2 (not 3)

Line 274 (Table 6) Aldrich (h is missing)

Author Response

Reviewer #2

The manuscript reports on interesting results on how the cyclodextrins dissolved in mouthwashes can influence the dissolution of nickel from dental alloys and its potential toxicity. It is recommended to publish after minor corrections.

1) It is suggested to cite the important recent review on the dental applications of cyclodextrins: Braga, S.S. Cyclodextrins as Multi-Functional Ingredients in Dentistry. Pharmaceutics 2023, 15, 2251. https://doi.org/10.3390/pharmaceutics15092251. 

The reference was added as number 24.

2) Give the water content of alpha- and gamma-cyclodextrin, too. (Table 6).

The water content of alpha- and gamma-cyclodextrins were added in Table 6 as suggested.

Some typos to correct:

Mind the font size (e.g. lines 88-95)

Line 77: mouthwashes and mouthwashes? (rephrase this sentence)

Line 133: Table 2 (not 3)

Line 274 (Table 6) Aldrich (h is missing)

We do apologize for the typos. We did our best to correct all of them throughout the ms.

Round 2

Reviewer 1 Report

Comments and Suggestions for Authors

I would like to thank the authors for sending answers to my questions. The authors fully answered my doubts. In its current form, the article is ready for publication.